# The Therapeutic Potential of Common Herbal and Nano-Based Herbal Formulations against Ovarian Cancer: New Insight into the Current Evidence

**DOI:** 10.3390/ph14121315

**Published:** 2021-12-17

**Authors:** Fatemeh Rezaei-Tazangi, Hossein Roghani-Shahraki, Mahdi Khorsand Ghaffari, Firoozeh Abolhasani Zadeh, Aynaz Boostan, Reza ArefNezhad, Hossein Motedayyen

**Affiliations:** 1Department of Anatomy, School of Medicine, Fasa University of Medical Sciences, Fasa 7345149573, Iran; f.rezaei67@yahoo.com; 2Student Research Committee, Shiraz University of Medical Sciences, Shiraz 1433671348, Iran; hossein.rgh.barca@gmail.com; 3Department of Physiology, School of Medicine, Shiraz University of Medical Sciences, Shiraz 1433671348, Iran; Khorsand2012@gmail.com; 4Department of Surgery, Faculty of Medicine, Kerman University of Medical Sciences, Kerman 7616913555, Iran; Firoozeh1981@gmail.com; 5Department of Obstetrics & Gynecology, Saveh Chamran Hospital, Saveh 3919676651, Iran; ainaz_bostan@yahoo.com; 6Department of Anatomy, School of Medicine, Shiraz University of Medical Sciences, Shiraz 1433671348, Iran; 7Autoimmune Diseases Research Center, Kashan University of Medical Sciences, Kashan 8715973474, Iran

**Keywords:** curcumin, quercetin, resveratrol, herbal medicine, nanotechnology

## Abstract

Ovarian cancer (OCa) is characterized as one of the common reasons for cancer-associated death in women globally. This gynecological disorder is chiefly named the “silent killer” due to lacking an association between disease manifestations in the early stages and OCa. Because of the disease recurrence and resistance to common therapies, discovering an effective therapeutic way against the disease is a challenge. According to documents, some popular herbal formulations, such as curcumin, quercetin, and resveratrol, can serve as an anti-cancer agent through different mechanisms. However, these herbal products may be accompanied by some pharmacological limitations, such as poor bioavailability, instability, and weak water solubility. On the contrary, using nano-based material, e.g., nanoparticles (NPs), micelles, liposomes, can significantly solve these limitations. Therefore, in the present study, we will summarize the anti-cancer aspects of these herbal and-nano-based herbal formulations with a focus on their mechanisms against OCa.

## 1. Introduction

Ovarian cancer (OCa) is characterized as the fifth most prevalent reason for death in women around the world because of its insidious initiation, weak prognosis, and rapid development. Based on estimations, annually, more than 100,000 females die due to OCa globally [1]. Mainly, OCa is categorized into three types: germ cell, sex-cord-stromal, and epithelia [2]. The most common form of OCa is epithelial OC (EOC), which is a heterogenic disorder, and histologically, EOC can be divided into four main subgroups, endometrioid, serous, mucinous, and clear cell carcinomas [3]. The etiology of OCa has not been completely illustrated yet, but it is shown that obesity, hereditary, aging, alcohol consumption, smoking, and diabetes mellitus are risk factors of OCa [4,5]. Plus, it is expressed that environmental, hormonal, and ovulation factors may have a role in the pathogenesis of OCa [6]. According to reports, OCa progression is linked with various pathways that interfered with the metabolism of energy, like galactose metabolism, which is related to the risk of OCa development [7]. OCa is mostly named the “silent killer” since the observed manifestations in the early stages of the disease are not clearly related to OCa [8]. The signs and symptoms of OCa can be general and ambiguous, such as abdominal pain, abnormal bowel habits, and early satiety [9]. There is a substantial need for using novel therapies for OCa due to disease recurrence and resistance to common therapies, such as chemotherapy and surgery [6,10,11]. In addition, these approaches can significantly along to cytotoxic impacts and severe complications [8]. Therefore, finding and using efficient curative methods against this gynecological tumor is indispensable. Among these, herbal remedies have obtained great attention from thousand years ago owing to their effectiveness against different ailments, such as cancer [12]. It is stated that some herbal compounds, like phenols, alkaloids, and lectins, can exert anti-cancer effects by apoptosis induction [13]. On the other hand, nano-based drug delivery systems, such as nanoparticles (NPs), nano micelles, liposomes, branched dendrimers, nanostructured lipid formulations, and nanocapsules, have been developed recently for the treatment of OCa [14]. Thus, the combination of herbal therapy and nano-based therapy may provide a new horizon in the improvement of OCa. For this reason, in this literature review, the therapeutic potential of some popular nano-based herbal formulations, namely *curcumin*, *quercetin*, and *resveratrol* [15,16,17], will be discussed.

## 2. Methods

In this literature review, we investigated accessible information from Google Scholar, PubMed, Scopus, Web of Science, Science direct, and Scientific Information Database until 2021. The MeSH terms used were: periodontitis, herbal medicine, herbal remedies, and natural products. According to the search strategy, 218 articles were discovered. After checking the titles, abstracts and manuscripts entirely cited, a collection of 126 papers were received and chosen according to the suitability indexes. The papers were performed around herbal medicine of different diseases especially ovarian cancer.

## 3. Ovarian Cancer and Its Pathogenesis

Presently, the pathogenesis and clinicopathological properties of OCa have not clearly been expressed [18]. However, some theories have been proposed concerning OCa origination (Figure 1) which includes (1) the gonadotropin theory characterizes over the induction of the epithelium of ovarian surface by hormonal receptors resulting in malignancy, (2) the continuous ovulation theory, in the location of which the cells of ovarian surface epithelium are damaged because of constant ovulation, (3) the origin cells for the majority of epithelial ovarian cancers are not derived from the ovary but mostly originated from the fallopian tube and develop to the ovary and more than it [19]. Regarding hormonal conditions, there is evidence that progesterone and androgens can elevate ovarian epithelium proliferation and subsequently lead to the formation of OCa. Indeed, the increment of androgens and estrogens taggers multiple pro-inflammatory agents resulting in immune activation. During ovulatory occurrences, a great number of molecules are produced, such as chemokines and cytokines, plasminogen activators, prostaglandins, interleukins, bioactive eicosanoids, tumor necrosis factors, collagenases, and a large number of growth factors and immune cells, which all trigger a pro-inflammatory occurrence. Such pro-inflammatory agents, like IL-8, CCL2/MCP-1, and CCL5/RANTES, are induced during each cycle of ovulation; therefore, the continuous ovulation theory recommends that the inflammation accompanied by other physiological situations potentiates OCa progression [20,21]. Plus, it is shown that IL-1β, IL-6, and TNF-α formed by activated immune agents and/or the tumor itself, induce the growth of cancer cells and affect the prognosis and clinical status of the disease via increasing resistance to chemotherapy and stimulating symptoms (e.g., weight loss, anemia, depression, and anorexia) [22]. Oxidative stress, namely reactive nitrogen species (RNS) and reactive oxygen species (ROS), are another factor involved in many pathological conditions, such as OCa, by genetic instability enhancement, angiogenesis promotion, and abnormality in cell proliferation [23,24]. Exogenous agents, like hypoxia, infection, and chronic inflammation, are among the main sources of oxidative stress [25]. Mounting evidence has demonstrated that ROS can modulate the biogenesis and expression of microRNAs by epigenetic alterations, regulating biogenesis course, and transcription factors [26]. MicroRNAs, as noncoding RNAs, have a role in chemoresistance, carcinogenesis, proliferation, apoptosis, cell cycle, invasion, and metastasis. Impairments of microRNAs can lead to the onset and progression of OCa [27]. Possibly, the most important feature of any cancer is genetic changes that mediate the development and progression of tumors [28]. In this line, it is revealed that the presence of mutations in *PTEN* (Phosphatase and tensin homolog), *P53*, *BRCA(Breast Cancer)1*, and *BRCA2* genes, and tumor suppressor factors can develop OCa (Figure 2) [29,30,31]. Endometrioid, serous, and mucinous types of OCa have shown mutations in *KRAS* (Kirsten rat sarcoma), *β-catenin*, *TFG-βRII*, and *BRAF* genes, which all are associated with the proliferation and cell growth processes [28]. 

## 4. Use of *Curcumin* and Its Nanoformulations against Ovarian Cancer

*Curcumin (CUR)* is characterized as a yellow and hydrophobic herbal component that is originated from the turmeric plant (*Curcuma longa L. Zingiberaceae*) [32]. Growing evidence has shown some positive effects of *CUR* in medicine, such as anti-tumor, anti-inflammatory, anti-oxidative, immunoregulatory, anti-fungus, and anti-bacterial features [33,34], such as breast, ovarian, prostate, gastric, colorectal, pancreatic, and cervical cancers [35,36,37,38]. Regarding anti-cancer mechanisms of *CUR*, it has to be said that several signaling pathways are affected by that, for example, JAK (Janus Activated Kinase)/STAT (signal transducer and activator), PI3K (Phosphoinositide 3-kinases)/Akt, MAPK (mitogen-activated protein kinase), NF-ĸB, p53, Wnt/β-catenin, and apoptosis-related signaling (Figure 3). Furthermore, CUR can suppress epithelial-mesenchymal transition (EMT), angiogenesis, proliferation, metastasis, and invasion of the tumor by modulating the expression of non-coding RNA (ncRNA) associated with the tumor [39,40,41,42]. In the study of Shi et al., it was revealed that CUR can considerably suppress the growth and stimulate apoptosis in human OCa cell line Ho-8910. In their research, the use of 40 μM *CUR* caused a reduction in pro-caspase-3, Bcl-X_L_, and Bcl-2, whereas Bax and p53 levels were elevated in the treated cells with *CUR* [38]. Triggering AMP-activated protein kinase (AMPK), which stimulates cell apoptosis and inhibits cell proliferation in several cancers, in a p38-dependent way is another mechanism of *CUR* action in ovarian cancer cells CaOV3 [43]. In an animal investigation on OCa, it was manifested that *CUR* can dramatically suppresses STAT3 and NF-ĸB signaling pathways [44]. Liu et al. have pointed out that *CUR* can stimulate human OCa cell autophagy through AKT/mTOR (mammalian target of rapamycin)/p70S6K pathway suppression [45]. Despite these, the clinical application of *CUR* has been limited owing to its instability and low water solubility, which in turn give rise to poor bioavailability of *CUR* in cancerous cells. Attempts toward elevating the therapeutic effectiveness of Cur have been carried out through various techniques [46]. For instance, drug delivery systems based on NP have attracted much attention. *CUR* can be encapsulated inside NPs to promote its water solubility, biocompatibility, and protection from breaking down, and enhance *CUR* accumulation in cancerous regions because of increased permeability and retention impact [47,48]. In the investigation of Xu et al., *CUR* was encapsulated with niosome and its therapeutic effects against OCa cells were assessed [49]. In the field of nanotechnology, niosomes are known as nonionic vesicles with a bilayer construction which have a high bioavailability and are a good candidate for drug delivery systems [49,50]. Xu and colleagues concluded that *CUR*-niosomes increase cytotoxic influences and induce apoptosis against OCa cells A2780 in comparison with free *CUR* [49]. An in vivo and in vitro investigation showed that nanocurcumin in combination with cisplatin, a common treatment for OCa, could lead to a remarked reduction of the weight and volume of ovarian tumors. In addition, this treatment decreased PI3K, JAK, TGF-β, Ki67 expression, and Akt phosphorylation [51]. Hu et al. (2020) demonstrated that the use of Docetaxel curcumin/methoxy poly (ethylene glycol)-poly (L-lactic acid) (MPEG-PLA) copolymers nanomicelles cause the Suppression of tumor proliferation and angiogenesis (Table 1). The study of Bondi et al. (2017) concluded that biocompatible Lipid nanoparticles as carriers improved curcumin efficacy in ovarian cancer treatment and caused the Reduction of cell colony survival, inhibition of tumor growth, and apoptosis induction (Table 1). In the study of Ghaderi et al. (2021) OCa cells were treated with free curcumin and Gemni-Cur the anticancer activity was investigated by uptake kinetics, cellular viability, and apoptotic assays. The results illustrate that Gemini-Cur nanoparticles have a great potential for developing novel therapeutics against ovarian cancer (Table 1). Previous studies also demonstrated that the use of curcumin and paclitaxel co-delivery by hyaluronic acid-modified drug-loaded polyethyleneimine and stearic acid caused Downregulation of P-glycoprotein, and suppression of tumor cell migration (Table 1). Dendrosomal nano-curcumin caused the reduction of cancer cell viability, a decrease of LncRNAs expression of H19 and HOTAIR, and an increase in the expression of MEG3 LncRNA and Bcl2 protein (Table 1). The study of Sandhiutami et al. (2021) showed that co-use of curcumin nanoparticles and Cisplatin caused the Decrease of ovarian tumor weight and volume, reduction of PI3K, TGF-β, JAK, and Ki67 expression, Akt and STAT3 phosphorylation, and decrease of IL-6 level (Table 1). In general, Curcumin is an efficient agent with anti-tumor, antioxidant, and anti-inflammatory activities. The main mechanisms of action by which curcumin exhibits its unique anticancer activity include inducing apoptosis and inhibiting proliferation and invasion of tumors by suppressing a variety of cellular signaling pathways.

## 5. Use of *Quercetin* and Its Nanoformulations against Ovarian Cancer

*Quercetin (Que)* is a polyphenolic compound present in various vegetables and fruits [64]. Several pharmacologic effects have been demonstrated for *Que*, such as anti-cancer, anti-proliferation, anti-inflammation, anti-oxidant, as well as anti-diabetes influences [65,66,67]. The growth of numerous cancers, like ovarian, colon, prostate, breast, cervical, gastric, and lung cancer, has been manifested to be diminished by *Que* [68,69,70,71,72,73]. Predominantly, the anti-cancer effects of *Que* are linked with the modulation of PI3K/Akt/ mTOR, STAT signaling pathway, expression of heat shock protein (HSP), intracellular pH modification, regulation of apoptosis-associated proteins, and the regulation of matrix metalloproteinases (MMPs), fibronectin, and vascular endothelial growth factor (VEGF) [74,75,76]. Apoptosis induction mechanisms have a key role in exerting anti-cancer impacts of *Que* possibly through the elevation of cytosolic Ca^2+^ levels, ROS generation, reduction of mitochondrial membrane potential, and surviving modulation [77,78,79,80]. *Que* also decreases the anti-apoptotic agents, for example, Bcl-2, Bcl-xL, whereas elevates the expression of pro-apoptotic agents, for instance, Bax, Bad, Bid, cyto-c, caspase-3, and caspase-9 (Figure 3) [81]. In vivo and in vitro studies have indicated that *Que* has a cytotoxic effect on OCa cells [67]. It is stated that *Que* can suppress the proliferation of OCa cells SKOV-3 in a dose-and time-dependent way. In addition, it can potentiate the apoptosis of these cell lines and attenuate the expression of survivin protein [82]. Regarding this disease, Liu and colleagues expressed that *Que* triggers autophagy by endoplasmic reticulum (ER) stress by the p-STAT3/Bcl-2 axis [83]. In this line, Yi et al. also highlighted that *Que* sensitizes human OCa cells to tumor necrosis factor-related apoptosis-inducing ligand (TRAIL), one of the strong anti-tumor agents in various cancer types [84,85,86]. They also observed that *Que* stimulated the expression of death receptor 5 (DR5) by JNK activation and CCAAT enhancer-binding protein homologous protein (CHOP) upregulation, while death receptor 4 (DR4) expression did not change by this phenol [84]. DR4 and DR5 belong to the TNF family and are induced through TRAIL, and CHOP is defined as a transcriptional factor that enhances apoptosis by the mediation of proportion of prosurvival Bcl-2 and the proapoptotic Bax [87,88,89]. Based on another work, *Que* stimulates radiosensitization via ATM phosphorylation induction and increases *p53* protein expression [90]. However, the pharmacological applications of quercetin are limited by its insolubility in water. Several approaches have been investigated to overcome these obstacles, such as the use of micelle, polymeric NPs, microemulsions, solid lipid NPs, liposomes, as well as liquid crystal systems [91,92,93]. It is stated that one of the ways to promote water solubility of hydrophobic medicinal compounds is encapsulation by polymer micelles [58]. Micelles are nanoscale colloidal aggregates obtained from amphiphilic surfactants, which their core and shell are hydrophobic and hydrophilic, respectively. These features make them a suitable carrier for hydrophobic drug delivery [94]. The line with this notion, Gao et al. encapsulated Que into the micelles of monomethoxy poly (ethylene glycol)-poly(3-caprolactone) (MPEG-PCL), these QU loaded MPEG-PCL (QU/MPEG-PCL) micelles with a drug loading of 6.9% had a mean particle size of 36 nm, rendering the complete dispersion of quercetin in water and they illustrated that intravenous injection of these micelles can significantly repress the growth of ovarian tumors by inducing the apoptosis of cancer cells and suppressing angiogenesis in vivo [58]. In the drug delivery system, also poly (3-caprolactone) and poly (ethylene glycol) (PCL/PEG) are block copolymers that are amphiphilic, biodegradable, and can easily be produced. [95,96]. Another work assessed the potential of PEGylated liposomal quercetin (Lipo-Que) in OCa cells in vivo and in vitro. Lipo-Que was prepared using a solid dispersion method, and the obtained Lipo-Que is monodisperse with a mean diameter of 163 ± 10 nm. They implicated that Lipo-Que suppresses the proliferation and growth, stimulates cell cycle arrest and apoptosis of ovarian tumors [60]. Liposomes are a drug delivery system that provides the possibility of administration of the lipophilic and hydrophilic drugs in a united formulation, and their outer membrane can be modified by the surface attachment with the PEG and/or other targeting molecules to boost their specificity [97]. Generally, quercetin is a desirable anticancer agent because of its natural origin, safety, and low cost relative to synthetic cancer drugs. Que decreases the expression of survivin protein, induces the expression DR5 and ATM phosphorylation, and increases p53 protein expression.

## 6. Use of *Resveratrol* and Its Nanoformulations against Ovarian Cancer

*Resveratrol (Res)* is defined as a non-flavonoid polyphenol compound possessing stilbene structural components, which are extensively found in lilies, grapes, and other herbs [63]. *Res* has been illustrated to have anti-tumor, anti-inflammatory, anti-oxidation, immunoregulatory, anti-virus, anti-microbial, neuroprotective, and anti-atherosclerosis influences [63,98,99]. Some documents revealed the positive effects of *Res* in different cancers, such as skin, ovarian, breast, colorectal, lung, and uterine cancer [100,101,102,103,104,105]. Several mechanisms are involved in exerting anti-tumor action of *Res*, for example, inflammation suppression through NLRP3 inflammasome inhibition, cyclooxygenase (COX) curbing, nuclear factor erythroid 2-related factor 2 (Nrf2) induction, and mitogen-activated protein (MAP) kinase phosphatase-1 (MKP-1) stimulation, which inhibits NF-ĸB pathway [99,106,107,108]. In OCa, the administration of *Res* curbs growth and stimulates cell death through apoptosome complex formation, caspase activation, and mitochondrial secretion of cytochrome c [109]. In the study of Kueck et al., *Res* suppressed glucose metabolism in OCa cells [110]. It seems that glycolysis, conversion of glucose into 3-carbon carbohydrates, and subsequently ATP formation, are needed for the enhancement of tumor growth [111]. Another study by Zhong and colleagues exhibited accumulated G1 phase, elevated apoptosis fraction, and simultaneous inhibition of STAT3, Notch, and Wnt signaling pathway (Figure 4) [112]. Some other investigations have manifested the anti-cancer effect of *Res* against OCa through AMPK activation, downregulation of the protein cyclin D1, EMT inhibition [113,114,115]. Regardless of the favorable results of Res in cancer treatment, its wide utilization has been limited because of its poor bioavailability, low solubility in water, instability, and unfavorable systemic delivery [116,117,118]. In contrast, nanotechnology-based strategies have been widely used to acquire promoted oral bioavailability, higher solubility, promoted solubility, and targeted release of Res [61]. In this line, Khatun et al. in their in vitro study used *Res*—(Zinc oxide) ZnO nanohybrid against OCa cell lines and demonstrated that this nanoformulation exerts anti-cancer effects by the generation of ROS [119]. ZnO NPs have attracted much attention due to their utility in cancer therapy and targeted drug delivery. In human cancer cells, these NPs can stimulate apoptosis by ROS generation, which in turn is associated with cellular apoptosis and DNA damage [119]. An in vivo investigation indicated that harnessing Res–BSA NPs attenuates tumor growth in nude mice with OCa through the induction of ovarian cancer cell necrosis and cellular apoptosis induction (Figure 4) [62]. BSA (bovine serum albumin) is a natural protein that is capable of the formation of complexes in different shapes. In addition to other beneficial features, BSA is nonimmunogenic, non-toxic, biodegradable, biocompatible. Thus, albumin particles can be a good candidate for drug delivery system [120]. In summary, resveratrol is a desirable substance with anti-cancer and anti-inflammatory activities. The anti-cancer effect of resveratrol is correlated with the damage of mitochondrial function that leads to increased ROS, apoptosis, downregulation of the protein cyclin D1 can fight against OCa. 

## 7. Conclusions

Recently, herbal remedy using some popular herbal spices, including CUR, Que, and Res has acquired much attention in the treatment of OCa, as one of the common gynecologic cancers, through different mechanisms. For example, CUR through suppression of EMT, angiogenesis, and STAT3 and NF-ĸB signaling, modulation of the expression of tumor-related-ncRNA, apoptosis stimulation, AMPK activation, inhibition of STAT3 and NF-ĸB signaling, and induction of autophagy can affect OCa. Que decreases the expression of survivin protein, induces the expression DR5 and ATM phosphorylation, and increases p53 protein expression. Res through mitochondrial secretion of cytochrome c, inhibition of glucose metabolism and STAT3, Notch, and Wnt signaling, and downregulation of the protein cyclin D1 can fight against OCa. However, these herbal products can have some negative aspects in terms of pharmacology, such as instability, poor bioavailability, and poor water solubility. Based on the evidence, using nano-based formulations from these herbal therapeutic candidates, for instance, gemini, ZnO nanohybrids, PEGylated liposome, NPs, micelles, niosome, not only can overcome these obstacles but also can improve the therapeutic potential of herbal medicine against OCa. However, more and larger researches are needed to show their therapeutic effects and mechanisms.

## Figures and Tables

**Figure 1 pharmaceuticals-14-01315-f001:**
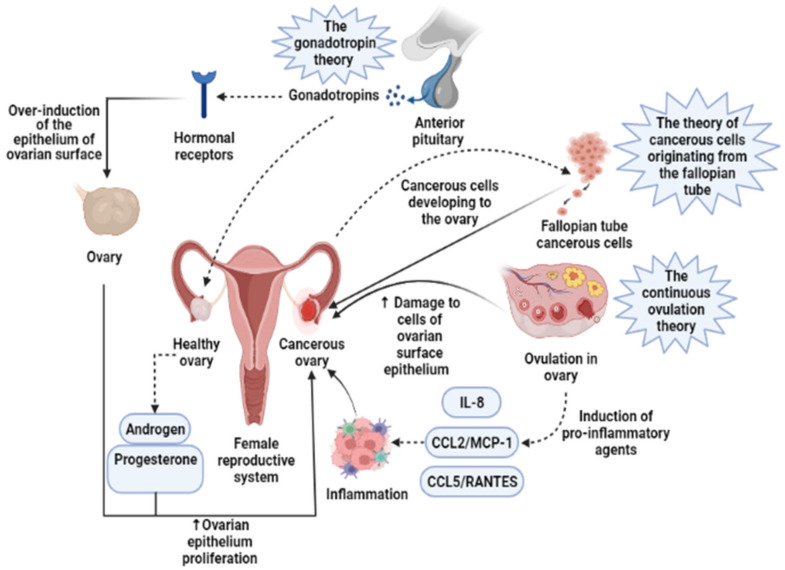
Three main theories regarding the development of ovarian cancer are based on induction of the epithelium of ovarian surface by hormonal receptors, increased induction of pro-inflammatory agents during continuous ovulation, and cancerous cells originating from the fallopian tube. IL-8, Interleukin-8; CCL2/MCP-1, Monocyte chemoattractant protein-1; CCL5/RANTES, CC Chemokine Ligand-5.

**Figure 2 pharmaceuticals-14-01315-f002:**
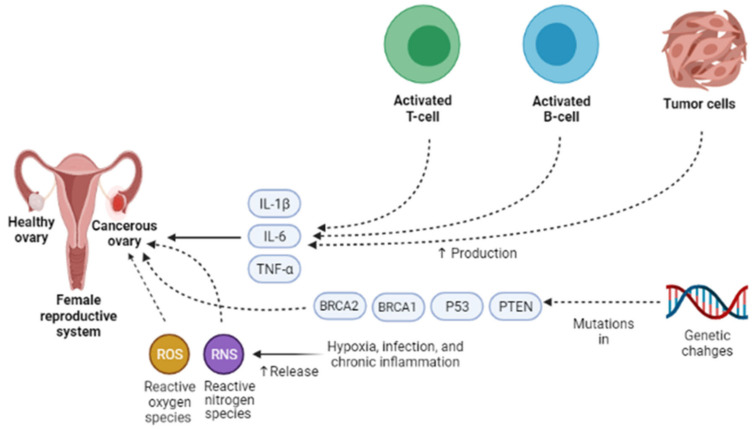
Different endogenous and exogenous factors modify the development and prognosis of ovarian cancer. IL-1β, IL-6, and TNF-α accelerate the growth of cancer cells and affect the prognosis and clinical status of the disease via increasing resistance to chemotherapy and stimulating symptoms. Exogenous factors, such as hypoxia, infection, and chronic inflammation, are the main sources of oxidative stress, namely reactive nitrogen species (RNS) and reactive oxygen species (ROS). They can contribute to ovarian cancer development via genetic instability enhancement, angiogenesis promotion, and abnormality in cell proliferation. One of the most important features of ovarian cancer is genetic changes that mediate the development and progression of tumors. In ovarian cancer, the presence of mutations in *PTEN*, *P53*, *BRCA1**,* and *BRCA2* genes, tumor suppressor factors, can lead to ovarian cancer development. IL-1β, Interleukin-1β; IL-6, Interleukin-6; TNF-α, Tumor necrosis factor α; PTEN, Phosphatase and tensin homolog; BRCA1, Breast cancer type 1; BRCA2, Breast cancer type 1.

**Figure 3 pharmaceuticals-14-01315-f003:**
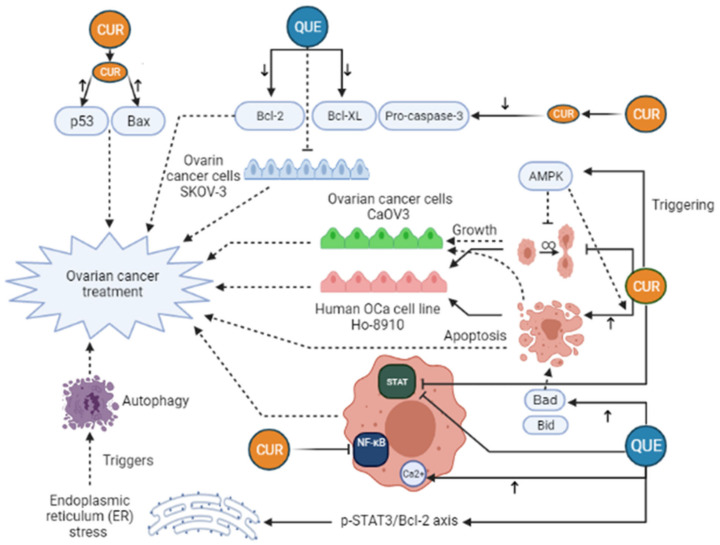
Curcumin (CUR) and Quercetin (Que) can exert an anti-cancerous effect on ovarian cancer in many different pathways. CUR triggers AMP-activated protein kinase (AMPK) that leads to stimulation of cell apoptosis and inhibition of cell proliferation. Moreover, CUR can decrease pro-caspase-3, Bcl-XL, and Bcl-2 levels, whereas Bax and p53 levels rise in the treated cells with CUR. These changes lead to ovarian cancer treatment. Furthermore, CUR can exert a significant inhibitory effect on STAT3 and NF-ĸB signaling pathways. Quercetin (Que) can modify many pathways and play a role in ovarian cancer treatment. Que decreases the anti-apoptotic agents, like Bcl-2, Bcl-xL, while it elevates the expression of pro-apoptotic agents, such as Bad and Bid, leading to increased apoptosis and ovarian cancer treatment. In addition, the elevation of cytosolic Ca^2+^ levels due to Que consumption can take part in ovarian cancer treatment. Que triggers autophagy by endoplasmic reticulum (ER) stress by the p-STAT3/Bcl-2 axis as well. Bcl-XL, B-cell lymphoma-extra-large; BAX, BCL2-associated X protein; Bcl-2, B-cell lymphoma 2; Bad, BCL2 associated agonist of cell death; Bid, BH3-interacting domain death agonist; STAT, Signal transducer and activator of transcription; NF-ĸB, Nuclear factor-kappaB.

**Figure 4 pharmaceuticals-14-01315-f004:**
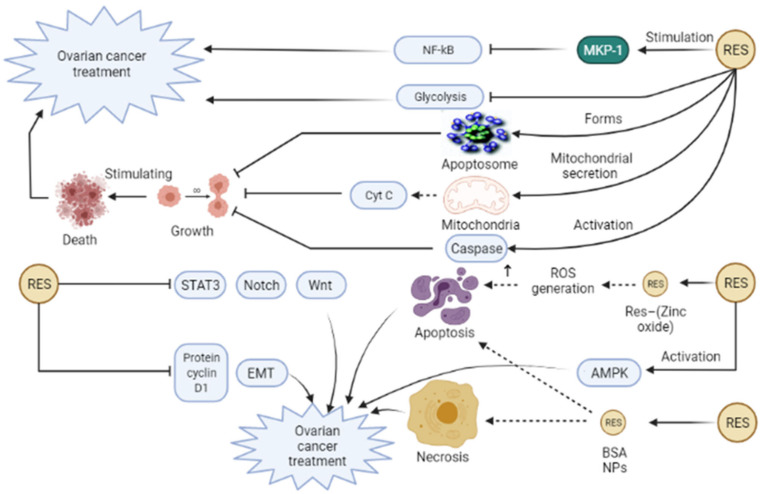
Resveratrol (Res) can trigger various mechanisms involved in ovarian cancer treatment. Res can stimulate mitogen-activated protein (MAP) kinase phosphatase-1 (MKP-1) that leads to inhibition of NF-ĸB pathway and subsequently contributes to inflammation suppression and ovarian cancer improvement. Furthermore, the administration of Res can stimulate apoptosome complex formation, caspase activation, and mitochondrial secretion of cytochrome c, which all end in inhibition of growth and stimulation of cell death. In addition, Res suppresses glycolysis in ovarian cancer cells that can be effective in ovarian cancer treatment. Res can act against ovarian cancer through AMPK activation, downregulation of the protein cyclin D1, and inhibition of EMT, STAT3, Notch, and Wnt signaling pathways, leading to ovarian cancer treatment. Moreover, consumption of Res—(Zinc oxide) ZnO nanohybrid can lead to the generation of ROS in ovarian cancer cell lines and exert anti-cancer effects on ovarian cancer. In addition, Res–bovine serum albumin (BSA) NPs induce ovarian cancer cell necrosis and cellular apoptosis, attenuating tumor growth in ovarian cancer. EMT, Epithelial–mesenchymal transition; Cyst C, Cytochrome c; STAT3, Signal transducer, and activator of transcription 3; NF-ĸB, Nuclear factor-kappaB.

**Table 1 pharmaceuticals-14-01315-t001:** Nano-based formulations of *curcumin*, *quercetin*, and *resveratrol* through various mechanisms affect ovarian cancer.

Type of Nano-Based Herbal Formulation	Mechanism/Effect	In Vivo/In Vitro	References
PLGA-phospholipid-PEG nanoparticles comprising *curcumin*	Downregulation of P-glycoprotein	In vitro	[52]
Niosome-encapsulated *curcumin*	Arresting the cell cycle at the S phase and apoptosis induction	In vitro	[49]
Docetaxel *curcumin*/methoxy poly (ethylene glycol)- poly (L-lactic acid) (MPEG-PLA) copolymers nanomicelles	Suppression of tumor proliferation and angiogenesis	In vivo/in vitro	[53]
*Curcumin*—loaded nanostructured lipid carrier	Reduction of cell colony survival, inhibition of tumor growth, and apoptosis induction	In vitro	[54]
Gemini *curcumin*	Apoptosis induction	In vitro	[55]
*Curcumin* and paclitaxel co-delivery by hyaluronic acid-modified drug-loaded polyethylenimine and stearic acid	Downregulation of P-glycoprotein, and suppression of tumor cell migration	In vivo/in vitro	[56]
Dendrosomal nano-*curcumin*	Reduction of cancer cell viability, decease of LncRNAs expression of H19 and HOTAIR, and increase in the expression of MEG3 LncRNA and Bcl2 protein	In vitro	[57]
Co-use of *curcumin* nanoparticles and Cisplatin	Decrease of ovarian tumor weight and volume, reduction of PI3K, TGF-β, JAK, and Ki67 expression, Akt and STAT3 phosphorylation, and decrease of IL-6 level	In vivo/in vitro	[51]
Encapsulated *quercetin* into monomethoxy poly (ethylene glycol)- poly (3-caprolactone)	Apoptosis induction and the suppression of angiogenesis	In vivo/in vitro	[58]
Encapsulated *quercetin* into methoxypoly(ethylene glycol) Poly(caprolactone)	Apoptosis induction and cell growth suppression	In vivo/in vitro	[59]
PEGylated liposomal *quercetin*	Apoptosis induction, cell proliferation inhibition, and arresting the cell cycle at G0/G1 and G2/M phases	In vivo/in vitro	[60]
*Resveratrol*—ZnO nanohybrid	Mitochondrial membrane depolarization and ROS formation	In vitro	[61]
RGD-conjugated *Resveratrol* human serum albumin nanoparticles	Reduction of cell viability and tumor growth inhibition	In vivo/in vitro	[62]
*Resveratrol*—bovine serum albumin nanoparticles	Reduction of cancer cell growth, activation of cytochrome C, upregulation of caspase-3 and caspase-3 expression	In vivo/in vitro	[63]

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
