# Peer review of "The Therapeutic Potential of Common Herbal and Nano-Based Herbal Formulations against Ovarian Cancer: New Insight into the Current Evidence"

_pharmaceuticals, 2021, doi:10.3390/ph14121315_

Round 1

Reviewer 1 Report

The review “The therapeutic potential of common herbal and nano-based herbal formulations against ovarian cancer: new insight into the current evidence” reports the latest research on the anti-cancer activity of curcumin, quercetin, and resveratrol formulated in nano-based systems with a focus on their mechanisms against ovarian cancer. The review is well written and organized. My recommendation is to describe the delivery nanosystem for each study cited, since Authors only say a few words on liposomes and micelles.

Author Response

Reviewer 1:

Hello Dear Editor

Thank you for your attention.

More details were given about the studies mentioned in the text. Liposomes and micelles were also described.

Reviewer 2 Report

The manuscript entitled “The therapeutic potential of common herbal and nano-based herbal formulations against ovarian cancer: new insight into the current evidence ”  described the use of  curcumin, quercetin, andresveratrol, as an anti-cancer agent against Ovarian cancer. Moreover the authors  summarized the anti-cancer aspects ofthese herbal and-nano-based herbal formulations  against Ovarian Cancer.The manuscript is interesting and the  presentation of the topic is legible. Thus, I do not have any points that should be improved.

Author Response

Reviewer 2:

Hello Dear Editor

Thank you for your attention.

Reviewer 3 Report

The article "The therapeutic potential of common herbal and nano-based herbal formulations against ovarian cancer: new insight into the current evidence" may offer some insight into treatment of the ovarian cancer. However, the article is relatively poorly structured and written. For example, the majority of the research that should constitute the text of the article is cited in Table 1. Thus, findings summarized in the Table 1 should be mentioned and elaborated in the text as well. The Table 1 itself should be mentioned in the text as well. In addition, Conclusions are general and do not summarize the main findings of the study. E.g. Conclusions section does not even mention the CUE, Que and Res nor their specific applications. Thus Conclusions should be completely re-written.
English in the article should be improved. Several examples of the English corrections and also other errors are given below, but the whole text should be carefully re-read and corrected, preferable by a native speaker. 

Line 47: "phenols, Alkaloids" please change to "phenols, alkaloids"
Line 55 "Ovaria" please change to "Ovarian"
Lines 57-63, Fig 1: Please reformulate and clarify the part of the text concerning OCa origination. It is not entirely clear in the present form.
Line  100: All plant names should be written in italic. Additionally, for each plant the plant family name should be mentioned on the first occurrence e.g. line 100 "Curcuma longa L. Zingiberaceae" where "Curcuma longa" should be written in italic.
Lines 103-104 "There are some clinical trials and literature studies indicating the curative potential of 103 CUR against several cancers," please change to "Scientific studies indicate the curative potential of CUR against several cancer types,"
line 128 "CUR in" please change to "CUR action in"
Line 136: please delete "to solving pure CUR limitations"
Line  136-137 "to the promotion of its" please change to "to promote its"
Line 138 "enhancement" please change to "enhance"
Lines 145, 177, 207, 251, 256, Table 1 "in vivo" "in vitro"  please change to italic
Since the chemical names have not be given for the other substances, please omit it in the case of quercetin too. In addition, the sentence in lines 162-163 should be simplified. Please change  "Quercetin (Que), chemically is called 3,30,40,5,7-pentahydroxy flavone, is named as a polyphenolic compound that is obtained from a variety of vegetables and fruits [52]." to  "Quercetin (Que)is a polyphenolic compound present in various vegetables and fruits [52]. 
The abbreviations should be defined at the first use (e.g. mTor in line 131 and not in the line 168). The other occurrences should be corrected as well (e.g. STAT etc.)
Lines 192-193 "In spite of beneficial effects of Que in medicine, it has limitations in the pharmacologic field in light of its insolubility in water."
please change to "However, the pharmacological applications of qurercetin are limited by its insolibility in water."
Line 209 The liposomes are described in the mid 20th century and can not be called novel systems thus "Liposomes are known as a novel drug delivery system" should be changed to "Liposomes are a drug delivery system"
Lines 209-212: Description of liposomes should be placed before the description of their use for the impovement of Que bioavailability (e.g. in line 206)
Line 211: Not all liposomes are pegylated. Thus, "is improved" should be changed to "can be modified". 
Line 241 "be" please change to "is"
Lines 258-262: Similar to liposomes, the characteristics of BSA should be mentioned before its use (e.g. in line 256)
Lines 258, 259 "BSA is described as a natural protein" please change to "BSA is a natural protein"
Line 260 "shapes and has several beneficial features, such as" please change to "shapes. In addition to other beneficial features, BSA is"
Lines 266-267 Cancers are treated and not the cancerous cells. Thus "many cancerous cells" should be changed to "many cancer types"
Lines 276-277: Only clinical trials may be taken as a final proof of the therapeutic usefulness. Thus the final sentence in Conclusions ("However, more researches with a large sample size on different nano-based herbal products are required to prove our results.") should be modified as follows "However, well conducted clinical trials with large number of patients are required to prove these hypotheses."

Author Response

Reviewer 3:

Hello Dear Editor

Thank you for your attention.

  • Findings summarized in the Table 1 mentioned and elaborated in the text. Also, Table 1 mentioned in the text
  • The conclusion section completely re-written.
  • Line 47: "phenols, Alkaloids" was changed to "phenols, alkaloids"
  • Line 55 "Ovaria" was changed to "Ovarian"
  • Fig 1 was corrected.
  • Line 100: The plant names was written in italic. For each plant the plant family name should be mentioned on the first occurrence e.g. line 100 "Curcuma longa L. Zingiberaceae" where "Curcuma longa" was written in italic.
  • Lines 103-104 "There are some clinical trials and literature studies indicating the curative potential of 103 CUR against several cancers," was changed to "Scientific studies indicate the curative potential of CUR against several cancer types,"
  • line 128 "CUR in" was changed to "CUR action in"
  • Line 136: "to solving pure CUR limitations" was deleted.
  • Line 136-137 "to the promotion of its" was changed to "to promote its"
  • Line 138 "enhancement" was changed to "enhance"
  • Lines 145, 177, 207, 251, 256, Table 1 "in vivo" "in vitro" was changed to italic
  • Tsentence in lines 162-163 was simplified. "Quercetin (Que), chemically is called 3,30,40,5,7-pentahydroxy flavone, is named as a polyphenolic compound that is obtained from a variety of vegetables and fruits [52]." Was changed to "Quercetin (Que)is a polyphenolic compound present in various vegetables and fruits [52].
  • The abbreviations was defined at the first use.
  • Lines 192-193 "In spite of beneficial effects of Que in medicine, it has limitations in the pharmacologic field in light of its insolubility in water."
    was changed to "However, the pharmacological applications of qurercetin are limited by its insolibility in water."
  • Line 209, "Liposomes are known as a novel drug delivery system" was changed to "Liposomes are a drug delivery system"
  • Lines 209-212: Description of liposomes was placed before the description of their use for the impovement of Que bioavailability
  • Line 211: "is improved" was changed to "can be modified".
  • Line 241 "be" was changed to "is"
  • Lines 258-262: the characteristics of BSA was mentioned before its use
  • Lines 258, 259 "BSA is described as a natural protein" was changed to "BSA is a natural protein"
  • Line 260 "shapes and has several beneficial features, such as" was changed to "shapes. In addition to other beneficial features, BSA is"

          The conclusion section was re-writen

Reviewer 4 Report

The manuscript can be published after major revision and my recommendations are:

  • please insert some personal comments regarding the reviewed papers.
  • please insert the methodology of choosing the cited papers (period of time, databases, etc);

Author Response

 Reviewer 4:

- Some personal comments regarding the reviewed papers was inserted.

- The methods section was added to text.

Round 2

Reviewer 3 Report

As the authors have accepted all the remarks, and corrected the manuscript accordingly, I have no further comments.

Reviewer 4 Report

This manuscript can be published in the journal. The authors improved the manuscript according to the recommendations.